# Structural changes in endocrine pancreas of male Wistar rats due to chronic cola drink consumption. Role of PDX-1

**Gabriel Cao**[1]◉*, **Julián González**[2]◉, **Juan P. Ortiz Fragola**[2]◉, **Angélica Muller**[2‡],
**Mariano Tumarkin**[2‡], **Marisa Moriondo**[2‡], **Francisco Azzato**[3‡], **Manuel Vazquez Blanco**[3‡],
**José Milei**[2,3]◉

**1** Centro de Altos Estudios en Ciencias Humanas y de La Salud (CAECIHS), Universidad Abierta
Interamericana, Buenos Aires, Argentina. Consejo Nacional de Investigaciones Científicas y Técnicas
(CONICET), Buenos Aires, Argentina, **2** Facultad de Medicina, CONICET, Universidad de Buenos Aires,
Instituto Alberto C. Taquini de Investigaciones en Medicina Traslacional (IATIMET), Buenos Aires, Argentina,
**3** Facultad de Medicina, Sexta Cátedra de Medicina, Hospital de Clínicas, Buenos Aires, Argentina

◉ These authors contributed equally to this work.
‡ These authors also contributed equally to this work
* gabrielcao@fibertel.com.ar

**Editor:** Muhammad Sajid Hamid Akash,
Government College University Faisalabad,
Pakistan, PAKISTAN

**Data Availability Statement:** All relevant data are
within the manuscript and its Supporting
Information files.

## Abstract

**Aim:** The objective of this work was to analyze the structural changes of the pancreatic
islets in rats, after 6 month consuming regular and light cola for 6 months. Also, we have
analyzed the possible role of PDX-1 in that process. Finally, with the available knowledge,
we propose a general working hypothesis that explains the succession of phenomena
observed. Previously, we reported evidence showing that chronic cola consumption in rats
impairs pancreatic metabolism of insulin and glucagon and produces some alterations typi-
cally observed in the metabolic syndrome, with an increase in oxidative stress. Of note It is
worth mentioning that no apoptosis nor proliferation of islet cells could be demonstrated. In
the present study, 36 male Wistar rats were divided into three groups to and given free
access to freely drink regular cola (C), light cola (L), or water (W, control). We assessed the
impact of the three different beverages in on glucose tolerance, lipid levels, creatinine levels
and immunohistochemical changes addressed for the expression of insulin, glucagon, PDX-
1 and NGN3 in islet cells, to evaluate the possible participation of PDX-1 in the changes
observed in α and β cells after 6 months of treatment. Moreover, we assessed by stereologi-
cal methods, the mean volume of islets (Vi) and three important variables: the fractional β
-cell area, the cross-sectional area of alpha (A α-cell) and beta cells (A β-cell), and the num-
ber of β and α cell per body weight. Data were analyzed by two-way ANOVA followed by
Bonferroni's multiple t-test or by Kruskal-Wallis test, then followed by Dunn's test (depend-
ing on distribution). Statistical significance was set at $p < 0.05$. Cola drinking caused impaired
glucose tolerance as well as fasting hyperglycemia (mean:148; CI:137–153; $p < 0.05$ vs W)
and an increase of in insulin immunolabeling (27.3±19.7; $p < 0.05$ vs W and L). Immunohisto-
chemical expression for PDX-1 was significantly high in C group compared to W (0.79±0.71;
$p < 0.05$). In this case, we observed cytoplasmatic and nuclear localization. Likewise, a mild
but significant decrease of in Vi was detected after 6 months in C compared to W group (8.2

**Funding:** The author JM, received specific funding for this work from University of Buenos Aires, Argentina, UBACYT 20020150100027BA.

**Competing interests:** The authors have declared that no competing interests exist.

±2.5; p<0.05). Also, we observed a significant decrease of in the fractional β cell area (78.2 ±30.9; p<0.05) compared to W. Accordingly, a reduced mean value of islet α and β cell number per body weight (0.05±0.02 and 0.08±0.04 respectively; both p<0.05) compared to W was detected. Interestingly, consumption of light cola increased the Vi (10.7±3.6; p<0.05) compared to W. In line with this, a decreased cross-sectional area of β-cells was observed after chronic consumption of both, regular (78.2±30.9; p<0.05) and light cola (110.5±24.3; p<0.05), compared to W. As for, NGN3, it was negative in all three groups. Our results support the idea that PDX-1 plays a key role in the dynamics of the pancreatic islets after chronic consumption of sweetened beverages. In this experimental model, the loss of islets cells might be attributed to autophagy, favored by the local metabolic conditions and oxidative stress.

## Introduction

Consumption of sugary beverages has been associated with an increase in all-cause mortality among older adults [1], primarily through an increase in cardiovascular disease associated mortality [2]. Currently, an estimated 1 in 6 deaths in the United States is attributed to coronary heart disease (CHD) [3]. In multiple experimental and long-term prospective studies, high consumption of dietary sugars, sugar-sweetened beverages (SSBs) in particular, has been associated with several CHD risk factors, including dyslipidemia [4], diabetes [5,7], and obesity [5–7].

Sugar-sweetened beverages are the single largest source of added sugar in the US diet[7]. Although consumption of SSBs has decreased in the past decade [8], data show a slight rebound in consumption in recent years among adults in most age groups. In developing countries, intake of SSBs is rising dramatically because of widespread urbanization and beverage marketing [9].

In previous studies, we reported evidence showing that chronic cola consumption in rats impairs pancreatic storage of insulin and glucagon and produces some alterations related to metabolic syndrome (hyperglycemia and hypertriglyceridemia) with an increase in oxidative stress [10–13]. Moreover, we also demonstrated that consumption of cola beverages, regardless of sugar content, increased the rate of atherosclerosis progression in ApoE2/2 mice, favoring aortic plaque enlargement (inward remodeling) over media thinning [14,15].

The pancreatic and duodenal homeobox 1 gene (Pdx1) encodes a transcription factor that critically regulates early pancreas formation and multiple aspects of mature beta cell function, including insulin secretion, mitochondrial metabolism, and cell survival. Reduced Pdx1 expression in the beta cell occurs in cellular models of glucose toxicity and accompanies the development of diabetes, correlating low Pdx1 levels with beta cell metabolic failure. Besides these aspects, little is known about the pathophysiologic process that justifies the changes observed in pancreatic islets by chronic consumption of SSBs in rats.

In this paper we focused the study on the effects in glucose and lipid homeostasis, pancreatic islets morphology and immunophenotype changes in alpha and beta cells observed after 6 months of chronic consumption of cola drink in male Wistar rats.

## Materials and methods

Animals were individually housed at the ININCA facilities (21±2˚C, at 12-h light-dark cycles 7am-7pm) and were fed a commercial chow (16%-18% protein, 0.2 g % sodium; Cooperación,

Buenos Aires, Argentina) *ad libitum*. Animal handling, maintenance and euthanasia procedures were performed according with international recommendations (Weatherall report, "The use of non-human primates in research.") [16]. The Committee of Ethics in Animal Research of the Instituto de Investigaciones Cardiológicas (ININCA) and the Institutional Animal Care and Use Committee (IACUC) of the Faculty of Medicine of the University of Buenos Aires (CICUAL, Institutional Committee for the Care and Use of Laboratory Animals) approved the study. For the euthanasia procedure an intraperitoneal injection of sodium pentobarbital was administered in a dose equivalent to nearly 3 times the anesthetic dose. We adopted the protocol proposed by Zatroch K.K. et al. (BMC Veterinary Research 2017; 13:60) with good results, which consists in the injection of 800 mg/kg of sodium pentobarbital in 3 ml of phosphate-buffered saline (PBS) performed in the right caudal quadrant of the abdomen. After 10 seconds, signs of ataxia appeared. At this point, the rats were placed in dorsal recumbency to evaluate the loss of righting reflex, which indicates loss of consciousness. The complete duration of the procedure was of at least 30 seconds [17].

## Animal care

The Wistar male rats employed in this experiment were provided by the Bioterio facilities of our Institution, thus reducing the animal's adaptive stress response, which is key since rats in experiments that include metabolic and hormonal interplays are quickly affected by this traumatic period. We designed a 7-day protocol consisting of 3 daily visits. The first 2 visits were dedicated to the control and measure of food and drink intake, general cleaning of the cages and replacement of the corn cob bedding. The third visit consisted in the clinical monitoring of the animals. According to the experimental protocol, the visits occasionally included blood extraction for biochemical analysis. Careful observation is the key to assessing the animal's wellbeing. In this sense, the discomfort could be manifested by reductions of liquid consumption, postural or fur changes, irritability, aggression, and decreased activity. In our protocol, the frequent animal monitoring explains our concern to minimize animal suffering. Also, animal wellbeing increases the confirmability of the experimental data.

## Experimental protocol

Thirty-six male Wistar rats were randomly distributed in 3 groups (12 animals per group), which were respectively assigned to different treatments according to beverage (as their only liquid source, *ad libitum*): W (water), regular cola (C) (commercially available sucrose-sweetened carbonated drink, Coca-Cola, Argentina) and light cola (L) (commercially available low-caloric aspartame-sweetened carbonated drink, Coca-Cola light, Argentina) for six months (end of treatment). Rats were weighed weekly. Food and drink consumption was assessed twice a week. Biochemical assays were performed at baseline; biochemical assays and histopathological data from autopsies were obtained at the end of treatment (6 months).

According to company specifications, Coca Cola$^{TM}$ is a carbonated water solution containing (approximate %): 10.6 g carbohydrates, sodium 7 mg, caffeine 11.5 mg, caramel, phosphoric acid, citric acid, vanilla extract, natural flavorings (orange, lemon, nutmeg, cinnamon, coriander, etc), lime juice and fluid extract of coca (*Erythroxylum novogranatense*). As far as nutritional information is concerned, the only difference between regular and light cola is the replacement of carbohydrates with non-nutritive sweeteners (aspartame + acesulfame K) in the latter [14]. Soft drinks had carbon dioxide content largely removed by vigorous stirring using a stirring plate and placing a magnetic bar in a container filled with the liquid prior to being offered to the animals at room temperature.

## Biochemical determinations

Plasma aliquots of blood collected from the tail vein after a 4-hour fasting period were used to measure the concentration of glucose and triglycerides by enzymatic colorimetric assays using commercially available kits (Sigma-Aldrich, USA) according to manufacturer's instruction [17]. An oral glucose tolerance test [18] was performed on each animal before and after treatment and plasmatic HDL and TG were also assessed. After treatment, plasmatic urea, creatinine, ASAT and ALAT were measured.

## Immunohistochemistry

For immunohistochemical techniques, additional sections were cut and mounted on positively charged slides. Alpha and beta pancreatic cells were evaluated in dewaxed sections using specific primary antibodies (mouse monoclonal anti-Glucagon antibody, clone 13D11.33, catalogue identification: MABN238, Lot.: 121M4805; mouse monoclonal anti-Insulin antibody, clone E11D7, catalogue identification: 05–1066, Lot.: 049K4761; Sigma-Aldrich Corp., St. Louis, Mo. USA). Rabbit polyclonal antibody against PDX1 (pancreatic and duodenal homebox 1, also known as insulin promoter factor 1 (Catalogue identification: ab47267, Lot.: GR313915-3; Abcam, Cambridge, UK)), was used to evaluate the glucose dependent insulin transcription and its Type 1 diabetes association. Rabbit polyclonal antibody against NGN3 or Neurogenin 3 (Catalogue identification: ab176124, Lot.: GR270873-3; Abcam, Cambridge, UK) was used to assess α and β cells expression of endocrine pancreas and its link to neurogenesis.

Before staining, sections were deparaffinized and incubated in 3% hydrogen peroxide for 10 minutes to quench endogenous peroxidase. After washing 3 times in PBS, nonspecific antibody binding sites were blocked with 4% skimmed milk and 1% bovine albumin in PBS. Sections were incubated with the primary antibodies diluted in blocking solution at 4˚C overnight. Negative controls were incubated with 4% skimmed milk and 1% bovine albumin in PBS. Sections were then washed 3 times in PBS and subsequently incubated for 60 minutes at room temperature, with the following biotinylated secondary antibodies: anti-Rabbit IgG (H+L) made in goat (Catalogue identification: BA-1000) Vector Laboratories, Inc. Origin: USA. Lot.: ZA0324, and Biotinylated Anti-Rabbit IgG (H+L) made in Goat (Catalogue identification: BA-1000) Vector Laboratories, Inc. Origin: USA. Lot.: ZA0324. Immunohistochemical staining was performed using a biotinylated-streptovidin-peroxidase complex (Dako Universal LSAB™+ Kit/HRP-K0690, Lot.: 10059904, Dako, Glostrup, Denmark) with DAB (3,3-Diaminobenzidin)-chromogen, Lot.: 10103067 (Dako-K3468, Dako, Glostrup, Denmark) as detection system according to manufacturer recommendations. Then, the sections were counterstained with Mayer's hematoxylin for the precise localization of the immunohistochemical expression.

## Morphologic analysis

At 6 months, necropsy was practiced. The whole pancreas was weighed and fixed in buffered 10% formaldehyde solution for 24 hours at room temperature. Later, the tissues were dehydrated in alcohols, cleared in xylene and embedded in paraffin. For light microscopy, a Nikon Eclipse 50i microscope (Nikon Corporation, Tokyo, Japan), equipped with a digital camera (Nikon Coolpix S4) and the Image-Pro Plus image processing software version 6.0 (Media Cybernetics, Silver Spring, Maryland, USA) were used. For stereological analysis, 3 μm width sections were cut from tissue blocks and stained with hematoxylin-eosin. Thus, digitalized images of 12 to 16 pancreatic islets per animal were obtained by systematic uniform random sampling of the tissue (total islets evaluated per group: 144–192). At this time, an orthogonal

grid with 50 test points representing a total area of $6.7 \times 10^4 \mu m^2$ (objective lens magnification: 40 X) was employed. Points were projected onto the fields of view and the number of points hitting the pancreatic islets was counted. In this way, the Cavalieri method with point-counting [19] was employed to estimate the mean volume of pancreatic islets:

$$\text{Vi} \ (10^4 \mu m^3) := t \ x \ A_p \ \sum_{i=0}^{n} Pi$$

$t$: mean section cut thickness; $A_p$: area associated with a point.
$P_i$: points counted on grid.

The evaluation of immunolabeled slides was carried out using a methodology that includes the transformation of the digitized images into the CMYK color scheme, using Adobe Photoshop v 21.0.2 software [20]. Subsequently, the images were separated into color channels, selecting the yellow channel image for analysis, previously converted to a gray scale image. Finally, with an appropriately calibrated image analysis software, the integrated optical density (IOD) of the immunolabelled insular areas was estimated. The IOD results from the product of the optical density (OD) and the immunolabeled tissular surface.

For glucagon:

$$glucagon \ (IOD) := \frac{OD \ for \ glucagon \ immunolabel}{area \ immunolabeled \ for \ glucagon}$$

For insulin:

$$insulin \ (IOD) := \frac{OD \ for \ insulin \ immunolabel}{area \ immunolabeled \ for \ insulin}$$

For TDX-1:

$$TDX - 1 \ (IOD) := \frac{OD \ for \ TDX - 1 \ immunolabel}{area \ immunolabeled \ for \ TDX - 1}$$

For NGN3:

$$NGN3 \ (IOD) := \frac{OD \ for \ NGN3 \ immunolabel}{area \ immunolabeled \ for \ NGN3}$$

The fractional beta cell area (Faβ) is an important and predictive parameter, given that their reduction is linearly related to insulin resistance [21–23]. Also, it is directly influenced by the beta cell mass changes. It is usually estimated by the quotient between the islet immunolabelled area for insulin and the pancreatic tissue surface, including the exocrine pancreas, both expressed in $\mu m^2$. From our point of view, it is more appropriate to use the area of the islet as denominator, excluding the pancreatic glandular component:

$$Fa\beta := \frac{area \ immunolabeled \ for \ insulin}{islet \ area} \times 100$$

Microscopically, there are no differences between the insular cell populations, therefore, the beta cells must be identified by anti-insulin antibodies immunolabelling. The area occupied by β cells can be estimated by a stereological technique, the point-counting, employing a previously defined test system in a microscopic, calibrated environment.

In this way, the Faβ value estimates the mean percentage of islet area occupied by β cells and expressed in percent (%).

Moreover, the number of alpha ($n\alpha$) and beta ($n\beta$) cells in the studying islets were estimated from the insular area immunolabeled for glucagon and insulin respectively, and the cross-sectional area of individual nucleated cell per islet. The cross-sectional area was estimated using the image processing software at 400X, regarding the cellular shape as spherical. We took the maximal diameter of each insular cell and calculated their area by applying the formula for the surface area of sphere. A sphere with radius r has a volume of:

$$V = \frac{4}{3}\pi r^3$$

then, their corresponding surface is:

$$A = 4\pi r^2$$

Given the relationship between body weight (BW) and $n\beta$ [22], we present this variable as the quotient between $n\alpha$, $n\beta$ and BW respectively ($n\alpha\ g^{-1}$ and $n\beta\ g^{-1}$):
For alpha cells:

$$n\alpha := \frac{area\ immunolabeled\ for\ glucagon}{cross-sectional\ area\ of\ individual\ nucleated\ cell\ per\ islet}$$

$$n\alpha\ g^{-1} := \frac{n\alpha}{BW}$$

Cross-sectional area of individual nucleated islet cell represents the area of alpha and beta cells employed to analyze the structural changes observed along the experiment.
For beta cells:

$$n\beta := \frac{area\ immunolabeled\ for\ insulin}{cross-sectional\ area\ of\ individual\ nucleated\ cell\ per\ islet}$$

$$n\beta\ g^{-1} := \frac{n\beta}{BW}$$

### Statistical analysis

Data were analyzed by two-way ANOVA followed by post-hoc tests (Bonferroni's multiple t-test or Kruskal-Wallis) followed by Dunn's test (depending on distribution) in order to evaluate the differences between the groups. Statistical significance was set at $p < 0.05$ and SPSS version 15.0 software was used to analyze data.

## Results

### Nutritional considerations

Regular cola drinking for 6 months caused an increase in liquid intake and a decrease in food intake ($p < 0.05$). Conversely, light cola drinking resulted in an increase in food and liquid intake ($p < 0.05$). No significant differences were observed in body weight (Table 1).

### Biochemistry

No significant differences were found at baseline (before randomization) between groups. Compared to control, regular cola drinking for 6 months produced alterations on lipid and glucose homeostasis and on renal function ($p < 0.05$). Glycemia was significantly higher in cola drinking rats both fasting at 30 and at 120 minutes (Table 2). On the other hand, HDL was

**Table 1. Nutritional aspects and plasmatic parameters after treatment.**

| Variable | W | C | L |
|---|---|---|---|
| Solid intake (g/kg/day) | 43±4 | 31±3.7* | 46±3.8* |
| Liquid intake (ml/kg/day) | 74.1±13.1 | 138±33* | 92.8±26.4* |
| Body weight at 6 months (g) | 119.7±3.1 | 118.8±4.4 | 116.8±9.7 |
| Creatinine (mg/dl) | 0.53(0.51–0.54) | 0.56(0.55–0.61)* | 0.6(0.55–0.63)* |
| ASAT (U/L) | 69(54–87) | 70(52–109) | 81(60–108) |
| ALAT (U/L) | 59(47–62) | 63(42–73) | 55(46–87) |

Treatment: Water (W), regular (C), or light cola drinking (L).

* $p < 0.05$ vs W.

significantly diminished. Triacylglycerols were markedly higher in the cola group, but it did not reach statistical significance, perhaps due to great data dispersion. Creatinine was also elevated in cola drinking rats ($p < 0.05$). Light cola drinking resulted in no significant changes in glycemia and lipid profile, but it increased creatinine ($p < 0.05$) (Table 2). No significant differences in ASAT and ALAT levels were observed in both, regular and light cola drinking groups (Table 1).

## Morphology and immunohistochemistry

After 6 months of treatment, C group showed significant increase in insulin IOD compared to W and L groups ($p < 0.0001$, respectively). Conversely, no significant differences were observed between L and W groups. What is more, cytoplasmatic expression for glucagon in C group was highest than W and L ($p < 0,0001$, respectively). Interestingly, L group showed the lowest glucagon IOD compared to W and C ($p < 0.0001$, respectively). Consumption of regular cola for 6 months significantly reduced the nuclear and/or cytoplasmic expression of PDX-1 in male Wistar rats, compared to W and L groups (both $p < 0.0001$). Similar values of PDX-1 IOD were obtained for W and L (see Table 3 and Fig 1).

Significant reduction of Vi was observed in C group, compared to W ($p < 0.0006$) and L ($p < 0.0001$). Conversely, L group showed a significant increase of Vi compared to W ($p < 0.0006$). Reductions of Faβ were seen in C and L groups compared with W ($p < 0.0001$, respectively). Likewise, a lower Faβ value was seen in C than L group ($p < 0.0001$). A similar profile was observed in C and L groups for $n\alpha \ g^{-1}$ and $n\beta \ g^{-1}$ values, compared to W group

**Table 2. Metabolic parameters before and after treatment.**

| Variable | Baseline | | | 6 months | | | Delta | | |
|---|---|---|---|---|---|---|---|---|---|
| | W Mean (CI) | C Mean (CI) | L Mean (CI) | W Mean (CI) | C Mean (CI) | L Mean (CI) | W Mean (CI) | C Mean (CI) | L Mean (CI) |
| Glu 1 (mg/dl) | 137(101–158) | 122.5(116–143) | 127(121–133) | 125(120–129) | 131(124–138)* | 125(116–129) | -12(-35-18) | 6(-8-14) | -7(-25-9) |
| Glu 2 (mg/dl) | 183(179–211) | 181(169–203) | 178(170–197) | 145(139–154) | 157(152–160)* | 136(128–143)* | -45(-63-25) | -31(-43—12.5)* | -45(-75-38) |
| Glu 3 (mg/dl) | 131(119–156) | 120(109–139) | 126(119–142) | 135(129–142) | 148(137–153)* | 131(121–133) | 0(-23-16) | 22(16–34)* | 4(-5-10) |
| HDL (mg/dl) | 21(17–27) | 22(21–24) | 21(18–21) | 33(30–36) | 25(22–31)* | 28(22–49) | 11(9–18) | 3,5(-0.5–9)* | 8(4–28) |
| TG (mg/dl) | 70(56–94) | 64(55–78) | 60(40–74) | 138(132–172) | 258(106–479) | 111(97–203) | 73(48–83) | 200(45–397) | 67(34–329) |

Delta: 6 months-baseline difference. Oral glucose tolerance test: Glu 1: basal glucose, Glu 2: glucose after 30 minutes. Glu 3: final glucose. HDL: high density lipoprotein. TG: triglycerides. Treatment: water (W), regular cola (C), or light cola drinking (L).

* $p < 0.05$ vs W.

**Table 3. Immunohistochemistry comparisons for insulin, glucagon and PDX-1 after 6 months of treatment.**

| Variable | W | C | L |
|---|---|---|---|
| Glucagon (IOD) | 5.8±3.1 (CI: 5.3–6.2) | 13.3±4.5 (CI: 12.5–14)*§ | 1.5±1.5 (CI: 1.3–1.8)* |
| Insulin (IOD) | 7.9±4.9 (CI: 7.2–8.8) | 27.3±19.7 (CI: 24.3–30.3)*§ | 16.8±19.9 (CI: 13.8–19.8) |
| PDX-1 (IOD) | 1.6±1.5 (CI: 1.4–1.8) | 0.79±0.71 (CI: 0.7–0.9)*§ | 1.4±1.1 (CI: 1.2–1.6) |
| Vi ($10^4$ µm³) | 9.6±0.7 (CI: 9.5–9.8) | 8.2±2.5 (CI: 7.9–8.6)*§ | 10.7±3.6 (CI: 10.2–11.2)* |
| A α-cell (µm²) | 93.1±27.2 (CI: 87.5–98.6) | 78.2±30.9 (CI: 72.1–84.2)*§ | 109.6±59.6 (CI: 96.7–122.6) |
| A β-cell (µm²) | 115.9±32.4 (CI: 109.7–122.0) | 78.2±30.9 (CI: 72.1–84.5)*§ | 110.5±24.3 (CI: 106.1–114.9) |
| Faβ (%) | 73.8±8.3 (CI: 72.7–74.8) | 38.4±12.1 (CI: 36.4–40.4)*§ | 46.8±6.4 (CI: 45.8–47.9)* |
| nα g⁻¹ | 0.2±0.04 (CI: 0.2–0.3) | 0.05±0.02 (CI: 0.050–0.055)*§ | 0.1±0.04 (CI: 0.10–0.12)* |
| nβ g⁻¹ | 0.2±0.04 (CI: 0.18–0.19) | 0.08±0.04 (CI: 0.07–0.08)*§ | 0.2±0.04 (CI: 0.14–0.15)* |

Also, we show stereological variables, such as: Vi ($10^4$ µm³) (mean pancreatic islet volume), A α-cell and A β-cell (cross-sectional area of α and β cells), Faβ (%) (fractional β cell area), nα g⁻¹ and nβ g⁻¹ (mean value of islet α and β cell number per BW). Treatment: water (W), regular cola (C), or light cola drinking (L).

* $p < 0.05$ vs W

§ $p < 0.05$ vs L.

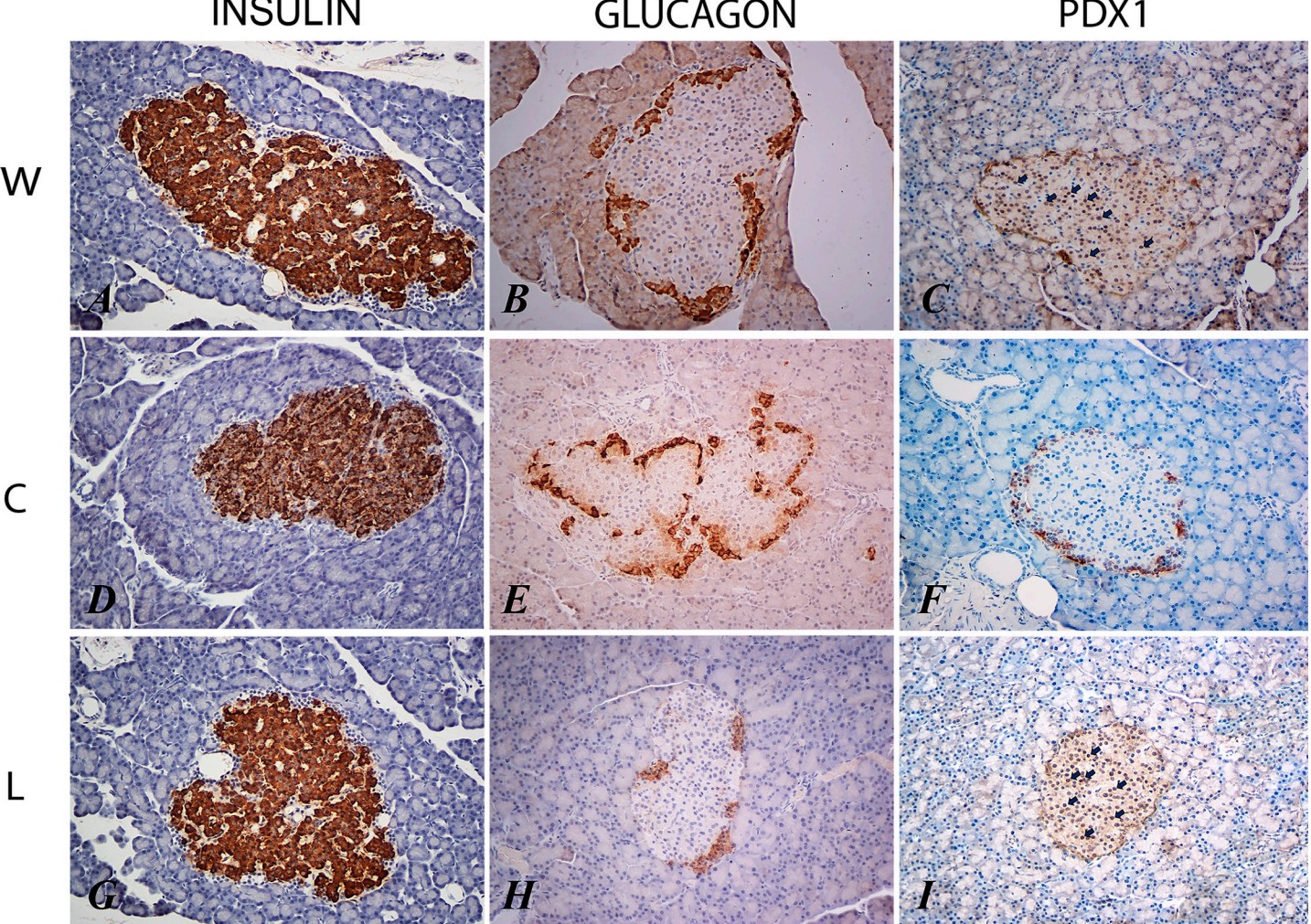

**Fig 1.** Immunostaining for insulin (A, D, G), glucagon (B, E, H) and PDX-1 (C, F, I) in water (W), regular cola (C), and light cola drinking (L) groups (see text for details). In C and I, black arrows indicate nuclear and cytoplasmatic immunostaining for PDX-1 (IHQ, 200X).

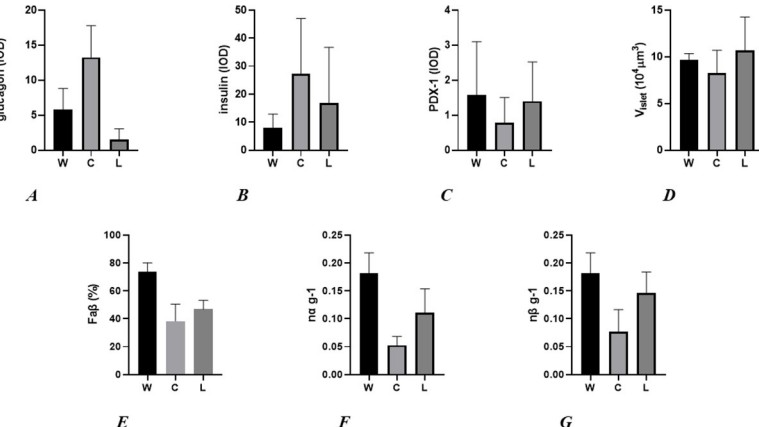

**Fig 2.** Graphics A, B, C: IOD for glucagon, insulin, and PDX-1 respectively; D–G: mean pancreatic islet volume, fractional β cell area and mean value of islet α and β cell number per BW in water drinking (W), regular cola drinking (C), and light cola drinking (L), respectively.

(p<0.0001, respectively). Also, C presented lower mean values than L group (p<0.0001, in both cases). Finally, NGN3 was negative in all three groups (see Table 3; Figs 2 and 3).

## Discussion

In this paper, we show that drinking cola for 6 months produced alterations on lipid and glucose homeostasis, in addition to changes in the structure and immunolabeling profile of pancreatic islets. Regular cola drinking caused an increase in liquid intake (empty calories) and a decrease in food intake, while light cola drinking resulted in an increase in food and liquid intake; none of these effects were associated with changes in body weight. These results are consistent with our previous studies [12,13,23] and studies by others [24–26]. The increase in fluid intake is partially explained by the sweet taste and sugar content rather than by increased thirst. This may be related to a "hedonic" effect caused by sugar ingestion. Excessive sugar consumption is associated with decreased striatal concentrations of dopamine [27]. On the contrary, acute effects of sugar consumption increase dopamine in the nucleus *accumbens*, an area closely related to reward feelings [28]. Some experiments with opioid-receptor knockout mice

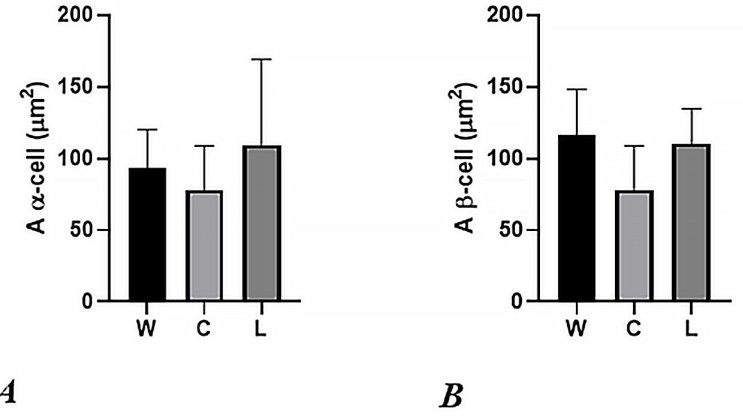

**Fig 3.** Graphics A and B represent the cross-sectional area of α and β cells respectively in water drinking (W), cola drinking (C), and light cola drinking (L) experimental groups.

suggest that opioid system may also be involved in reward sensation related to sugar intake [29,30]. In connection to this, behavioral studies in mice showed a preference for caloric liquids independently of sweet taste [31].

The effect on solid food consumption of regular and light cola drinks may be related also to reward. The effect of sucrose on appetite is complex. In a mice study, animals given a premeal load of sugar consumed more chow. In the same study, sugar seemed to decrease the content of neuropeptide Y and agouty-related protein in the arcuate nucleus, both with orexigenic effect, immediately after sugar ingestion, but then showed to cause a sort of rebound effect with an increase of both neurotransmitters just before chow ingestion [32]. Besides, the effect of glucose and fructose (both components of sucrose) seem to be opposite. While glucose leads to an increase in the hypothalamus content of cholecystokinin (satiety-inducing) [33] and decreased ghrelin (orexigenic) [31], fructose does the contrary [30,32]. Furthermore, both chronic consumption of glucose and fructose decrease polypeptide YY serum levels and hypothalamic proopiomelanocortin mRNA expression, both with satiety-inducing effects [34]. Taking all these data into account, sugar should prompt an increase in food intake, but the opposite is observed quite consistently in the many studies cited [10,12,13,23–25,32,33,35].

The role of caffeine has also been a matter of debate in some cola drinking studies. Caffeine has shown to enhance thermogenesis and energy expenditure in rats [36], but doses used in this paper are far higher than doses we calculated to be provided by cola and light cola drinking [12]. Consistently with other studies, we found that regular cola impaired glucose tolerance and decreased HDL cholesterol [10–12,24]. This hyperglycemic effect is explained by several mechanisms, including impairment of the expression of adipocytokines, altered hepatic glucose management with increased glycogenolysis, gluconeogenesis and pancreatic dysfunction [2,10].

Regular cola drinking also induced an increase in triacylglycerols levels. This can be ascribed to the excess of calories delivered by sugar content of cola beverages and especially to fructose content. Glucose enters glycolytic pathway and is metabolized into pyruvate, which in turn is converted to acetyl-CoA, which then enters de tricarboxylic acid cycle. When glucose is abundant in hepatocytes, the tricarboxylic acid cycle saturates, and excess of acetyl-CoA is derived to fatty acid synthesis and then to triacylglycerols synthesis. Fructose enters the hepatocyte and is phosphorylated in carbon 1, bypassing the limiting enzyme of glycolysis, phosphofructokinase-1[40]. That explains the relation between fructose consumption and fatty liver disease [40].

After 6 month of regular cola consumption, pancreatic islets showed a significant immunolabeling increase for insulin in both C and L groups. In the light cola group, this phenomenon might be explained by two mechanisms: 1- acesulfame K and other non-nutritive sweeteners can activate incretin release and stimulate pancreatic insulin production [37,38], and 2- phenylalanine, partially metabolized to aspartame, can synergistically stimulate insulin secretion [38–40]. This mechanism could be justifying the glycemia decrease observed in the glucose tolerance test in light cola drinkers (improved glucose tolerance).

In this paper we analyzed the effect of regular cola and light cola as a whole industrial product. We consider that caramel contributes to the caloric content of sugar-sweetened beverages, added to what is provided by sucrose. In this sense, we assume that both components increase glycemic values in rats and oxidative stress in pancreatic islets.

Longtime consumption of regular cola as unique source of liquid induced a significant increase in glucagon immunolabeling compared with W group. However, in the same period, consumption of cola light suppressed the immunolabeling of glucagon to values lower than W and C groups. In rats, mechanisms associated to stimulation or inhibition of glucagon release includes hyperglycemia and hypertriglyceridemia, respectively. Some reports show that blood

glucose concentration of 7 to 8 mmol/L, maximally inhibit glucagon secretion [41,42]. Interestingly, concentrations above 20 mmol/L stimulate glucagon secretion in mice [43]. This phenomenon was associated with variation of cytosolic levels of $Ca^{2+}$. Also, hypertriglyceridemia and increased blood fatty acid levels reduce glucagon release in rats, dogs and guinea pigs [44]. Likewise, chronic consumption of acesulfame K inhibit glucagon synthesis by stimulation of glucagon-like peptide-1 (GLP-1), an incretin produced in the gut [45,46]. Also, long-time consumption of acesulfame K and aspartame increase blood levels of triglycerides, ASAT and ALAT transaminases, creatinine and urea, reducing blood levels of HDL cholesterol [46]. Finally, in C group high production of insulin hypothetically could inhibit release of glucagon through the production of somatostatin from δ-cells. Nevertheless, low levels of glucagon persist, suggesting that hyperglycemia possesses greater metabolic influence on α-cells (Fig 1).

Chronic consumption of regular cola, reduced the mean islet Vi, associated to increased immunolabeling for insulin, compared to W group. This stereological change could be explained by the reduction in area and number of β-cells, involving an increased synthesis and release of insulin by each individual cell. This finding suggests the development of metabolic stress. In previous papers [10,13], we reported an increased expression of insular thioredoxin-1 and peroxiredoxin-1 in cola drinking rats, suggesting a key participation of oxidative stress in the pathogenic changes observed in pancreatic islets. This scenario inhibits PDX-1 activity and critically suppresses the cellular capability of hypertrophy, hyperplasia, and protection of β-cells [47–49]. PDX-1 is a transcription factor that plays an important role in pancreas development, β-cell differentiation and maintenance of mature β-cells function by regulating several related genes [50]. Chronic hyperglycemia promotes oxidative stress in β-cells, activating several signal transduction pathways, including c-Jun N-terminal kinase (JNK). In addition, JNK overexpression decreases PDX-1 to DNA binding, reducing insulin gene expression and secretion [51]. In previous papers [10,13], we tested the participation of apoptosis and proliferation related to the structural changes observed in pancreatic islets, by immunostaining for caspase 3 and PCNA [10]. Low expression associated to scarce insular PDX-1 activity was observed, especially in C group. Also, oxidative stress induces the translocation of PDX-1 from nuclei to cytoplasm, via the activation of JNK transduction pathway, justifying the cytoplasmic localization observed by immunohistochemical studies [52–54]. The changes described above lead to a significant reduction in Faβ, indicative of peripheral insulin resistance. Persistence of this anomaly maintains insulin "hyperproduction" worsening the paracrine autoregulation of the pancreatic islet cells, resulting finally in islet "dysregulation", frequently seen in type 2 diabetic patients [55].

In this point, we note the relation between oxidative stress, reductions in Vi, cross-sectional area, number of β-cells per body weight and the lower proliferative and apoptotic activity. Accumulation of reactive oxygen species (ROS) and reactive nitrogen species (RNS) being among the main intracellular signal transducers sustaining autophagy [56]. We postulate that longtime consumption of regular cola in rats increases oxidative stress in pancreatic islets, inducing cellular loss (probably via autophagy, a mechanism independent of caspase cascade). In turn, size reduction of β-cells in C group compared with W group could be explained by the elevated metabolic demands, the low levels of ATP and the cellular need to obtain molecules and energy, which is obtained from own cellular components.

Consumption of light cola for 6 months leads to a modest but significant increase in Vi, compared to W and C groups. Conversely, nα g$^{-1}$ proved a more reduced value than W group, but to minor severity than nβ g$^{-1}$. Interestingly, compared to W group, a non-significant increased value of cross-sectional area for α-cells was seen. (Fig 3). Although we accept the dispersion of the values, we clearly noted that the value of cross-sectional area of α-cells was increased, suggesting a possible common factor that, also, favored the increased value observed

in Vi, given that we have not seen changes in the cross-sectional area of β-cells after chronic consumption of light cola, compared to W group (Fig 3). In a previous paper, our group could not demonstrate significant proliferative activity in pancreatic islet cells [10]. This finding suggests that the increase of cross-sectional area of α-cells is related to hypertrophic phenomena, associated to a significant reduction in glucagon IOD, possibly mediated by PDX-1 activity. Along this line, Gao T et al [57] described that PDX-1 activates important genes for β-cell identity, suppressing those associated to α-cell phenotype. In L group, variables such as Vi, $n\alpha\ g^{-1}$, $n\beta\ g^{-1}$, A α-cell and A β-cell (Figs 2 and 3) showed a similar profile to C group, but with minor severity. These findings could be related to the minor caloric delivery by light cola, contrarily to regular cola. In this sense, long-term consumption of light cola could potentially induce similar metabolic disturbances to type 2 diabetes.

The transcriptional factor neurogenin 3 (NGN3) is involved in differentiation, transdifferentiation and regeneration of pancreatic islet cells [58]. We could not demonstrate NGN3 activation in pancreatic islets after chronic consumption of regular cola. Then, and in opposition to our previous hypothesis [10], transdifferentiation seems not to be the predominant phenomenon that explains the dynamics of pancreatic islet cells observed in this model.

Finally, we took account of the paper by Ferretti F and Mariani M [59] that showed the consumption of sugar-sweetened beverages (SSB) per geographic region in 2015. In Argentina, 157.40 liters/person/year of SSB were consumed, equivalent to 78.7 liters in 6 months (182.5 days). In our experiment, the rats in C group (regular cola drinkers) consumed 138±33 ml/kg/day, equivalent to 16.39 ml per day (see Table 1):

$$[(118.8 \pm 4.4\ g\ x\ 138 \pm 33\ ml)/1000\ g] = 16.39\ ml\ per\ day$$

In 6 months: 16.39 ml x 182.5 days: = 3000 ml or 3 liters.

Then, in 6 months, consumption by rats is 26.2 times smaller than the 78.7 liters consumed by humans.

## Conclusions

Based on the previous and current findings, we postulate a hypothesis about the pathophysiology of this experimental model. No significant differences were observed in body weight after chronic consumption (6 month) of regular and light cola as unique sources of liquid. The finding was associated to increase in glycemia and lipidic disturbances, especially after regular cola drinking. Under similar conditions, our group demonstrated that cola drinking induces islet and systemic oxidative stress in rats by the reduction of plasma levels of antioxidants α-tocopherol and ubiquinone-10 ($CoQ_{10}$), increased immunohistochemical expression of Trx1 and Prx1, respectively, and reductions in the HOMA-IR index [10,11]. These phenomena could explain the reduction of α and β-cell mass ($n\alpha\ g^{-1}$, $n\beta\ g^{-1}$, respectively) in addition to Faβ. Another work by us showed that aerobic exercise was protective for pancreatic islets [13]. Moreover, Saisho Y et al. postulated that a reduction in Faβ (and the β-cell mass) by approximately 50% in adult non-human primates, reduces the immediate secretable insulin pool, favoring insulin resistance [60]. At a pancreatic islet level, accumulations of ROS and RNS damage β-cells mass, through PDX-1 activity suppression, promoting autophagia and consequent cellular loss. In this respect, β-cell promotes an increased compensatory synthesis, storage, and release of insulin. This scenario favors metabolic stress and lack of ATP, accelerates cellular loss and islet dysfunction.

This pathophysiologic mechanism explains the pancreatic islet changes observed after 6 month of regular cola consumption (Figs 2 and 3). As for light cola, long-term consumption may develop a similar mechanism than regular cola, but for a longer time, due to the lowest caloric intake that it provides.

Our results suggest, for the first time, that PDX-1 plays a key role in the dynamics of the pancreatic islets after chronic consumption of sweetened beverages. The loss of islets cells might be attributed to autophagy, favored by the local metabolic conditions (Figs 2 and 3).

## Study limitations

The experiment's duration poses an important limitation in relation to the structural and metabolic progression of the pancreatic islet injury. In this sense, we are planning on conducting an experiment employing male Wistar rats consuming regular cola and light coca for 12 months, as their unique source of liquid, to evaluate the initial changes that lead to metabolic syndrome, especially at a structural level. In this point, the introduction of ultrastructural analysis by transmission electron microscopy will allow to explore the role of the autophagia in the reduction in islets Vi observed in C group after regular cola consumption for 6 months. In addition, we employed PCR and/or western blot for the detailed analysis of PDX-1 regulation. Also, since mTOR is a central regulator of autophagy [61], we plan to use rapamycin, an inhibitor of mTOR, for the study of the role of autophagy during the evolution of type 2 diabetes in this experimental model.

## Supporting information

**S1 File. Laboratory data.**
(XLSX)

**S2 File. Nutritional aspects and plasmatic parameters.**
(XLSX)

**S3 File. Immunohistochemistry.**
(XLSX)

**S4 File. Stereological variables.**
(XLSX)

## Acknowledgments

We especially want to acknowledge Nora Paglia, VMR, for her professional support and dedicated supervision of this experiment and animal protect.

## Author Contributions

**Conceptualization:** Gabriel Cao, Francisco Azzato, Manuel Vazquez Blanco, José Milei.

**Data curation:** Juan P. Ortiz Fragola, Angélica Muller, Mariano Tumarkin.

**Investigation:** Julián González.

**Methodology:** Gabriel Cao, Julián González, Angélica Muller, Mariano Tumarkin, Marisa Moriondo.

**Project administration:** Julián González, Angélica Muller.

**Resources:** Mariano Tumarkin, Marisa Moriondo.

**Software:** Gabriel Cao.

**Supervision:** Francisco Azzato, Manuel Vazquez Blanco, José Milei.

**Validation:** Gabriel Cao.

**Visualization:** Gabriel Cao, Juan P. Ortiz Fragola, Mariano Tumarkin.

**Writing – original draft:** Gabriel Cao, José Milei.

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
