## [Decision Letter · Decision Letter 0]

13 Jan 2021

PONE-D-20-36227

Could PDX-1 explain the changes observed in endocrine pancreas due to chronic cola drink consumption in rats? A global point of view

PLOS ONE

Dear Dr. Cao,

Thank you for submitting your manuscript to PLOS ONE. After careful consideration, we feel that it has merit but does not fully meet PLOS ONE’s publication criteria as it currently stands. Therefore, we invite you to submit a revised version of the manuscript that addresses the points raised during the review process.

Present study is unique in its nature as it is a new breakdown in diabetic studies that links to pancreatic morphological alterations especially in relation to gene. But the reviewers have criticized on the data. Moreover, I also found that experimental evidences are also required to support the histological results of PDX-1. Language of this paper needs to be critically revised to remove grammatical mistakes and syntax errors.

We look forward to receiving your revised manuscript.

Kind regards,

Muhammad Sajid Hamid Akash

Academic Editor

PLOS ONE

Journal Requirements:

2 Please ensure you have discussed any potential limitations of your study in the Discussion.

3.At this time, we request that you  please report additional details in your Methods section regarding animal care, as per our editorial guidelines:

(1) Please state the source of mice used in the study

(2) Please describe the  care received by the animals, including the frequency of monitoring and the criteria used to assess animal health and well-being.

Thank you for your attention to these requests.

4. Please ensure you have provided the  product number and any lot numbers of all the antibodies purchased from for your study.

5. Please include your statement regarding the method of euthanasia in the manuscript Methods.

6. Please amend either the abstract on the online submission form (via Edit Submission) or the abstract in the manuscript so that they are identical.

Reviewers' comments:

Reviewer's Responses to Questions

**Comments to the Author**

1. Is the manuscript technically sound, and do the data support the conclusions?

Reviewer #1: Partly

Reviewer #2: Partly

2. Has the statistical analysis been performed appropriately and rigorously? 

Reviewer #1: Yes

Reviewer #2: Yes

3. Have the authors made all data underlying the findings in their manuscript fully available?

Reviewer #1: Yes

Reviewer #2: Yes

4. Is the manuscript presented in an intelligible fashion and written in standard English?

Reviewer #1: Yes

Reviewer #2: Yes

5. Review Comments to the Author

Reviewer #1: The current study is good, deep and unique in its nature. It is a new breakdown in diabetic studies that links to pancreatic morphological alterations especially in relation to gene. The idea of this pharmacogentic type research can be proved as a guide towards new establishments and findings for the treatments of such type of chronic ailments which are playing more crucial role in human mortalities.

This manuscript describes genetic effects on insulin resistance in diabetic studies. This is a novel field of research. Different studies have been carried out on diabetes and diabetes related complications as well as disorders but no study is designed and conducted to correlate genetic makeup with insulin resistance with having regular drinks in daily life.

Thank you for inviting me as reviewer to review this manuscript. This study is really good and the paper is well written and structured. However, in my opinion, the paper has some short comings in regards to some data analysis and text and these certain issues should be addressed before publishing this kind of research. Refer to comments Please for details.

Specific Comments

Manuscript has several flaws as follows:

Title of Manuscript

Tile of manuscript that has been sent through mail was "Could PDX-1 explain the changes observed in endocrine pancreas due to chronic cola drink consumption in Rats? A global point of view " but the manuscript titled with "Could PDX-1 explain the changes observed in endocrine pancreas due to chronic cola drinks consumption in Rats? A global point of view".or "Role of TDX-1 in the endocrine pancreas of Rats after chronic consumption of cola drinks ". If focusing is on PDX-1 ,then please maintain the same at every required place of paper.

ABSTRACT

1. Please describe aim/objective of the study before discussing the methedology of research study.

2. No inferential statistics reported in results. please add the type of statistical methods i. e; one way ANOVA or any other method of analyzing the data before discussing the results.

3. Please add briefly the statistical values including especially p-values and then highlight the results of the present work.

4. Again the word TDX has been used.

KEY WORDS

1. Here also TDX-1is used in place of PDX-1.

METHODS

1. Large groups of animals have been used. It maybe considered as wastage of resources especially of animals as well as money

2. Due to large groups, a big data is collected and might be produce some errors in calculation of different values for different parameters

.

3. In morphological analysis, to calculate Fab, self supposed formula has been used. Do you think, this is a proper way to produce and analyze the findings. ( from mathematical point of view)

4. Which anaesthesia is used for animals, their doses and for how long?????????

RESULTS

1. Figures 1,2 and 3 should have expectations along with them. Not should be discussed on apart from their images.

DISCUSSION

1. In this part, there is detailed discussion about different aspects but the focusing point PDX-1 (insulin promoter factor) is not deeply discussed. More considerations should be there to get better understanding about that explains different changes in pancreas.

2. For "Hedonic effect " no references are provided.

3. According to ref 46, there should be increased levels of ASAT, ALAT, Creatinine and TGs but this shows no significant changes in ASAT and ALAT levels. Might be the reason is large distributed data and applications of statistics in improper way for that or if any other explanations are there????

4. Repetition of paragraphs (consumption of light cola................ Type 2 diabetes). Please do correct.

5. Any justifications for ( doses used in this paper are far higher than doses calculated to be provided cola drink consumption.)????

6. In my opinion, caramel/ caramel colour is also a part of cola drinks and it can have destructive role for pancreatic morphology. No discussions or considerations about this in this article, are there. It also should be a focusing point.

CONCLUSIONS/FURTHER RECOMMENDATIONS

1. Collected data of this research is Good, but more satisfactory and well explained information can be obtained from further investigations for this gene.

2. In introduction, it is mentioned that PDX-1 encodes a transcription factor that critically regulates early pancreas formations and multiple aspects of mature beta cells function" early pancreas development indicates early maturation and development of pancreatic cells. So, if this types of study is carried out in neonatal or especially gestational period, then the more clear information and data could be there to explain the changes related to PDX-1 gene.

LIMITATIONS OF STUDY

1. Please add limitations of the study.

2. The limitations can be added as a separate heading that may be just replace the funding information into limitations of the study.

Reviewer #2: Comments to the Author

This manuscript nice presented and interesting finding to assess the impact of the three different beverages in glucose tolerance, lipid levels, creatinine levels, and immunohistochemical changes addressed for the expression of insulin, glucagon, PDX-1 and NGN3 in islet cells, to evaluate the possible participation of PDX-1 in the changes observed in α and β cells after 6 months of treatment. Overall, these findings are important and interesting. However, further, improvement is necessary to solidify the Manuscript.

Here are a few comments and questions:

1. In the current manuscript, the author used the title Could “TDX-1 explain the changes observed in the endocrine pancreas due to chronic cola drink consumption in rats? A global point of view. Revised the title. The title does not completely match with the study.

2. To solidify the idea of the PDX-1 current study, the author should add more experiments like real-time PCR result or western blot results.

3. There is some typing error in table:2

4. In the current study what the effect of chronic or light cola on liver function test after 6 months. Like ALT, AST, ALP Bilirubin, etc.

5. Confusion in the group of chronic cola and light cola was only given cola for drinking.?

Authors address these deficiencies, then the manuscript should be considered for publication.

6. PLOS authors have the option to publish the peer review history of their article (what does this mean?). If published, this will include your full peer review and any attached files.

Reviewer #1: **Yes: **Musaddique hussain

Reviewer #2: **Yes: **DR. MUHAMMAD TARIQ

---

## [Author Response · Author response to Decision Letter 0]

24 May 2021

Response to Reviewers

Academic Editor:

1) “…I also found that experimental evidence is also required to support the histological results of PDX-1”.

Answer: We consider that there is sufficient evidence to postulate the hypothesis raised in the text of the paper, and presented below:

Chronic consumption of regular cola, reduced the mean islet Vi, associated to increased immunolabeling for insulin, compared to W group. This stereological change could be explained by reduction of area and number of β-cells, involving an increased synthesis and release of insulin by each individual cell. This finding suggests developing of metabolic stress. In previous papers [10,13] we reported an increased expression of insular thioredoxin-1 and peroxiredoxin-1 in cola drinking rats, suggesting a key participation of oxidative stress in the pathogenic changes observed in pancreatic islets. This scenario inhibits PDX-1 activity and critically suppress the cellular capability for hypertrophy, hyperplasia and protection of β-cells [47-49]. In previous papers [10,13] we tested the participation of apoptosis and proliferation related to the structural changes observed in pancreatic islets, by immunostaining for caspase 3 and PCNA [10]. Low expression was seen, associated to scarce insular PDX-1 activity, especially in C group. Also, oxidative stress induces the nucleo-cytoplasmic translocation of PDX-1, justifying the cytoplasmic localization observed by immunohistochemical studies [50]. The changes described above lead to significant reduction of Faβ, indicative of peripheral insulin resistance. Persistence of this anomaly, maintains insulin “hyperproduction” worsening the paracrine autoregulation of the pancreatic islet cells, conducing finally to “dysregulation” of this structure, condition frequently seen in type 2 diabetic patients [51].

In this point, we note the logical relation between oxidative stress, reductions of Vi, cross-sectional area, number of β-cells per body weight and the lowest proliferative and apoptotic activity. Accumulation of reactive oxygen species (ROS) and reactive nitrogen species (RNS) being among the main intracellular signal transducers sustaining autophagy [52]. We postulate that long time consumption of regular cola in rats increase oxidative stress in pancreatic islets, inducing cellular loss (probably via autophagy, a mechanism independent of caspase cascade). In turn, size reduction of β-cells in C group compared with W group, could be explained by the elevated metabolic demands, the low levels of ATP and the cellular need to obtain molecules and energy, which is obtained from own cellular components. 

The consumption of light cola for 6 months leads a modest but significantly increase of Vi, compared to W and C groups. Conversely, nα g-1 proved a reduced value than W group, but to minor severity than nβ g-1. Interestingly, compared to W group, a non-significantly increased value of cross-sectional area for α-cells was seen. (Fig. 3). Although we accept the dispersion of the values, we clearly noted that the value of cross-sectional area of α-cells was increased, suggesting a possible common factor that, also, favored the increased value observed in Vi, given that we not seen changes in the cross-sectional area of β-cells after chronic consumption of cola light, compared to W group (Fig. 3). In a previous paper, our group could not demonstrate a significative proliferative activity in pancreatic islet cells [10]. This finding suggests that the increasing of cross-sectional area of α-cells are related to hypertrophic phenomena, associated to significantly reduction of glucagon IOD, possibly mediated by PDX-1 activity. In this line, Gao T et al [53] described that PDX-1 activate important genes for βcell identity, suppressing those associated to α-cell phenotype. In L group, variables such as Vi, nα g-1, nβ g-1, A α cell and A β cell (Fig. 2 and 3) showed similar profile than C group, but with minor severity. These finding could be related to the minor caloric delivery by light cola, contrarily to regular cola. In this sense, long term consumption of light cola, potentially could produce similar metabolic disturbances than type 2 diabetes.

2) Please ensure you have discussed any potential limitations of your study in the Discussion.

Answer: The paper has some limitations. One of that is the need of prolonged of experimental time, since, although groups C and L (especially the first) show morphological and functional signs of evolution towards type 2 diabetes, it has not yet been clearly demonstrated. The purpose is to continue with the detailed study of this model, initially extending its duration for 12 months. In this line, is also interesting dilucidated the pathophysiological evolution of the rats chronically consuming light cola. Another limitation is the scarce information obtained about the intimal mechanisms related to the beta and alpha cells dynamics. In a previous work [10], our group could not prove a active participation of the proliferative or apoptotic activities on changes observed in beta cell population.

3) At this time, we request that you please report additional details in your Methods section regarding animal care, as per our editorial guidelines:

(A) Please state the source of mice used in the study.

(B) Please describe the care received by the animals, including the frequency of monitoring and the criteria used to assess animal health and well-being.

Answer (A and B): The Wistar male rats, employed in this experiment, were provided by the Bioterio facilities of our Institution, reducing the adaptative stress period of the animal’s, an important situation since the experiments that include metabolic and hormonal interplays are quickly affected by this traumatic period. We planned a protocol of 7 days, consisting in 3 daily visits, 2 making different activities: control and measure of food and drink consumption, general clean of the cages and replace of the corn cob bedding. The third visit consisted in the clinical monitoring of the animals. According to the experimental protocol. occasionally the visits included blood extraction for biochemical analysis. The careful observation is the key to detect the presence of animal wellbeing. In this sense, the discomfort could be manifested by reductions of liquid consumption, postural or fur changes, irritability, aggressivity and reductions of activity. In our protocol, the frequent animal monitoring explains the concern to minimize animal suffering. Also, animal wellbeing increases the confirmability of the experimental data.

4) Please ensure you have provided the product number and any lot numbers of all the antibodies purchased from for your study.

Answer: 

A) Primary antibodies employed in the study:

-Monoclonal Anti-Glucagon Antibody produced in Mouse. (Catalogue identification: G2654). Origen: Israel. Lot.: 121M4805 (Sigma-Aldrich Corp., St. Louis, Mo. USA).

-Monoclonal Anti-Insulin Antibody produced in Mouse. (Catalogue identification: I2018). Origen: Israel. Lot.: 049K4761 (Sigma-Aldrich Corp., St. Louis, Mo. USA).

-Rabbit polyclonal antibody against PDX1, (pancreatic and duodenal homebox 1 also known as insulin promoter factor 1). Origen: Cambridge. Lot.: GR313915-3 (Catalogue identification: ab47267) (Abcam, Cambridge, UK).

-Rabbit polyclonal antibody against NGN3 or Neurogenin 3. Origen: Cambridge. Lot.: GR270873-3 (Catalogue identification: ab176124) (Abcam, Cambridge, UK).

B) Secondary antibodies employed in the study:

-Biotinylated Anti-Rabbit IgG (H+L) made in Goat. (Catalogue identification: BA-1000). Vector Laboratories, Inc. Origen: USA. Lot.: ZA0324.

-Biotinylated Anti-Mouse IgG (H+L) made in Goat. (Catalogue identification: BA-9200). Vector Laboratories, Inc. Origen: USA. Lot.: ZE1207.

C) Other products employed:

-LSAB+System HRP. Biotinylated Link universal. Streptavidin+HRP. Dako. Origen: USA. Lot.: 10059904.

-Liquid DAB+Substrate Chromogen System. DAB+Chromogen/DAB+ Substrate Buffer. Dako. Origen: USA. Lot.: 10103067.

5) Please include your statement regarding the method of euthanasia in the manuscript Methods.

 Answer: The euthanasia procedure was performed employed intraperitoneal injection of sodium pentobarbital, in one dose equivalent to, for almost, 3 times the anesthetic dose. We adopt the protocol proposed by Zatroch K.K. et al. (BMC Veterinary Research 2017;13:60) with good results. Consists in the injection of 800 mg/kg sodium pentobarbital in 3 ml of phosphate-buffered saline (PBS) performed in the right caudal quadrant of the abdomen. After 10 seconds, signs of ataxia appear. In this point, the rats were placed in dorsal recumbency to evaluate the loss of righting reflex, sign of consciousness loss. Finally, the complete duration of the procedure was at least of 30 seconds.

6) Please amend either the abstract on the online submission form (via Edit Submission) or the abstract in the manuscript so that they are identical.

Answer: The abstract was corrected.

Reviewer #1: Specific Comments

1) Title of Manuscript

Tile of manuscript that has been sent through mail was "Could PDX-1 explain the changes observed in endocrine pancreas due to chronic cola drink consumption in Rats? A global point of view " but the manuscript titled with "Could PDX-1 explain the changes observed in endocrine pancreas due to chronic cola drinks consumption in Rats? A global point of view”. or "Role of TDX-1 in the endocrine pancreas of Rats after chronic consumption of cola drinks ". If focusing is on PDX-1, then please maintain the same at every required place of paper.

Answer: The Manuscript Title was changed, following the Reviewer #1 recommendations: “Structural changes in endocrine pancreas of male Wistar rats due to chronic cola drink consumption. Role of PDX-1”.

Also, the denomination "TDX-1" was replaced by the correct expression "PDX-1", a repeated involuntary error. For this reason, we express our sincere apologies.

2) ABSTRACT

A. Please describe aim/objective of the study before discussing the methodology of research study.

Answer: The suggestion was added.

B. No inferential statistics reported in results. Please add the type of statistical methods i. e; one way ANOVA or any other method of analyzing the data before discussing the results.

Answer: The suggestion was added.

C. Please add briefly the statistical values including especially p-values and then highlight the results of the present work.

Answer: The suggestion was added.

D. Again, the word TDX has been used.

Answer: Corrected.

3) KEY WORDS

A. Here also TDX-1is used in place of PDX-1.

Answer: Corrected.

4) METHODS

A. Large groups of animals have been used. It may be considered as wastage of resources especially of animals as well as money.

Answer: We appreciate the comment, however, the number of experimental units per group is correct. Always is necessary maintain a balance between the number of animals and the used resources, often scarce. A statistical power of 80% according to tables was fixed in the experimental design and, therefore, the "n" per experimental group. This was due to the importance of the hypothesis proposed for this model, after inductive-deductive work, and raised as a pathophysiological sequence that could potentially cause type 2 diabetes. For the other hand, the morphological variables presented in this work, are quantitative and continuous, for that they have been statistically analyzed with a non-parametric ANOVA test, such as the Kruskal-Wallis test. Also, a single value per animal and per group was obtained, therefore, we did not agree the idea of "big data". In this sense, many data obtained for the same quantitative continuous variable, tends to reduce her standard deviation, acquire a Gaussian distribution. In this condition, should be evaluated the use of the parametric ANOVA test. In addition, this condition potentially could demonstrate minimal differences between experimental groups, that not always are relevant since a biological point of view. These situations did not happen in the present work.

B. Due to large groups, a big data is collected and might be produce some errors in calculation of different values for different parameters.

Answer: Was answered in the previous topic.

C. In morphological analysis, to calculate Fab, self-supposed formula has been used. Do you think, this is a proper way to produce and analyze the findings? (from mathematical point of view).

Answer: Along the time, different research groups presented papers where studied the mechanisms related to the type 2 diabetes. Also, they included distinct morphological aspects estimated by the Faβ formula, which is mathematically logical. Microscopically, not exists differences between the different insular cell populations, therefore, the beta cells must be identified by anti-insulin antibodies immunolabelling. The area occupied by beta cells, can be estimated by a stereological technic, the point-counting, with a previously defined test system, in a microscopic, calibrated environment and expressed in µm2. Also, the islet area is estimated too by the same method and expressed in µm2. The quotient between the insulin immunolabeled area and the islet area are multiplied by 100, resulting a percentage value that correspond to the fractional area of beta cells. That is, Faβ value estimate the percentage of islet area occupied by beta cells. 

D. Which anesthesia is used for animals, their doses and for how long?????????

Answer: Was answered previously.

5) RESULTS

A. Figures 1,2 and 3 should have expectations along with them. Not should be discussed on apart from their images.

Answer: I believe that those figures, presents adjusted and concise descriptions, and then facilitate the interpretation of data and morphological changes.

6) DISCUSSION

A. In this part, there is detailed discussion about different aspects but the focusing point PDX-1 (insulin promoter factor) is not deeply discussed. More considerations should be there to get better understanding about that explains different changes in pancreas.

Answer: We add more aspects about PDX-1.

B. For "Hedonic effect " no references are provided.

Answer: References were already provided [30-33].

C. According to ref 46, there should be increased levels of ASAT, ALAT, Creatinine and TGs but this shows no significant changes in ASAT and ALAT levels. Might be the reason is large distributed data and applications of statistics in improper way for that or if any other explanations are there????

Answer: In the reference 46 (actually ref 49 due the inclusion of more bibliography), Cong W et al. do not comment alterations referred to ASAT, ALAT and Creatinine. In the case of TGs, the authors show not significantly differences between the experimental groups. In our work, although C group showed high levels of TGs, it did not show significant differences compared to W and L groups.

D. Repetition of paragraphs (consumption of light cola................ Type 2 diabetes). Please do correct.

Answer: Corrected.

E. Any justifications for (doses used in this paper are far higher than doses calculated to be provided cola drink consumption)????

Answer: Ferretti F and Mariani M showed the consumption of sugar-sweetened beverages (SSB) per geographic region in 2015. In Argentina, 157.40 liters/person/year of SSB were consumed, equivalent to 78.7 liters in 6 months (182.5 days). In our experiment, the rats of C group (regular cola drinkers) consumed 138±33 ml/kg/day, equivalent to 16.39 ml per day (see Table 1): 

[(118.8±4.4 g x 138±33 ml) / 1000 g] := 16.39 ml per day

In 6 months: 16.39 ml x 182.5 days := 3000 ml or 3 liters

Then, in 6 months, correspond to 26.2 times smaller than 78.7 liters consumed by humans.

Ferretti F and Mariani M. Sugar-sweetened beverage affordability and the prevalence of overweight and obesity in a cross section of countries. Global Health. 2019; 15: 30. doi: 10.1186/s12992-019-0474-x.

(The text and the reference were included in the manuscript).

F. In my opinion, caramel/ caramel colour is also a part of cola drinks and it can have destructive role for pancreatic morphology. No discussions or considerations about this in this article, are there. It also should be a focusing point.

Answer: in this paper, we analyzed the effect of regular cola and light cola such industrial product in toto. Also, we consider that caramel contribute to caloric content of the sugar-sweetened beverages, added to provide by sucrose.

(The text was included in the manuscript).

7) CONCLUSIONS/FURTHER RECOMMENDATIONS

A. Collected data of this research is Good, but more satisfactory and well explained information can be obtained from further investigations for this gene.

Answer: The experimental duration results an important limit for the structural and metabolic progression of the pancreatic islet injury. In this sense, we are planning develop an experiment employing male Wistar rats consuming regular cola and coca light for 12 months, as unique source of liquid, to evaluate the initial changes that lead to metabolic syndrome, specially at structural levels. In this point, the introduction of ultrastructural analysis by transmission electron microscopy will allow to explore the role of the autophagia in reduction of islets Vi observed in C group after regular cola consumption for 6 months. In addition, we employ PCR and/or western blot for the detailed analysis of PDX-1 regulation. Also, such as mTOR is a central regulator of autophagy [61], we plan to use rapamycin, an inhibitor of mTOR, for the study of autophagy role during the evolution of type 2 diabetes in this experimental model.

B. In introduction, it is mentioned that PDX-1 encodes a transcription factor that critically regulates early pancreas formations and multiple aspects of mature beta cells function" early pancreas development indicates early maturation and development of pancreatic cells. So, if this type of study is carried out in neonatal or especially gestational period, then the more clear information and data could be there to explain the changes related to PDX-1 gene.

Answer: Some experiments [50] shows pancreatic agenesia in mice homozygous for a targeted mutation to PDX-1. In the other hand, in my opinion, are difficult generate experimental conditions that provoke oxidative stress into pancreatic islet level during the gestational period, given that this condition globally affect the embryonic and fetal development. Also, generate hyperglycemia previously to gestational period imply work with pregnant rats. In that conditions, the animals generally demonstrate fertility difficulties. However, result interesting an experimental design during the neonatal period, including chronic consumption of sugar-sweetened beverages as unique source of liquid, considering one experimental group supplemented by essential vitamins and mineral while another without this supplementation.

Reviewer #2: Comments to the Author

This manuscript nice presented and interesting finding to assess the impact of the three different beverages in glucose tolerance, lipid levels, creatinine levels, and immunohistochemical changes addressed for the expression of insulin, glucagon, PDX-1 and NGN3 in islet cells, to evaluate the possible participation of PDX-1 in the changes observed in α and β cells after 6 months of treatment. Overall, these findings are important and interesting. However, further, improvement is necessary to solidify the Manuscript.

Here are a few comments and questions:

1. In the current manuscript, the author used the title Could “TDX-1 explain the changes observed in the endocrine pancreas due to chronic cola drink consumption in rats? A global point of view. Revised the title. The title does not completely match with the study.

Answer: The title was change to match with the study.

2. To solidify the idea of the PDX-1 current study, the author should add more experiments like real-time PCR result or western blot results.

Answer: We are planning develop an experiment employing male Wistar rats consuming regular cola and coca light for 12 months, as unique source of liquid, to evaluate the initial changes that lead to metabolic syndrome, specially at structural levels. In this point, the introduction of ultrastructural analysis by transmission electron microscopy will allow to explore the role of the autophagia in reduction of islets Vi observed in C group after regular cola consumption for 6 months. In addition, we employ PCR and/or western blot for the detailed analysis of PDX-1 regulation. Also, such as mTOR is a central regulator of autophagy [61], we plan to use rapamycin, an inhibitor of mTOR, for the study of autophagy role during the evolution of type 2 diabetes in this experimental model.

This text was included in the manuscript.

3. There is some typing error in table: 2.

Answer: The table 2 was corrected.

4. In the current study what the effect of chronic or light cola on liver function test after 6 months. Like ALT, AST, ALP Bilirubin, etc.

Answer: In cola drinking rats, HDL was significantly diminished and triacylglycerols were markedly higher in the cola group, but it did not reach statistical significance, perhaps due to great data dispersion. On the other hand, light cola drinking resulted in no significant changes in lipid profile. Finally, no significant differences in ASAT and ALAT levels were observed in both, regular and light cola drinking groups.

5. Confusion in the group of chronic cola and light cola was only given cola for drinking.?

Authors address these deficiencies, then the manuscript should be considered for publication.

Answer: In this experiment, were employed three experimental groups assigned to different treatments according to beverage, W (water), regular cola (C) (commercially available sucrose-sweetened carbonated drink) and light cola (L) (commercially available low-caloric aspartame-sweetened carbonated drink). C and L groups chronically (6 months) consumed regular and light cola, respectively, as unique font of liquid.

Not confusion was possible, because the rats were housed in separate cages and different colors for the conserving bottles were employed (red for regular cola and black for light cola).

---

## [Editor Report · Decision Letter 1]

28 May 2021

Structural changes in endocrine pancreas of male Wistar rats due to chronic cola drink consumption. Role of PDX-1.

PONE-D-20-36227R1

Dear Dr. Cao,

We’re pleased to inform you that your manuscript has been judged scientifically suitable for publication and will be formally accepted for publication once it meets all outstanding technical requirements.

Kind regards,

Muhammad Sajid Hamid Akash

Academic Editor

PLOS ONE
---

## [Editor Report · Acceptance letter]

2 Jun 2021

PONE-D-20-36227R1 

Structural changes in endocrine pancreas of male Wistar rats due to chronic cola drink consumption. Role of PDX-1. 

Dear Dr. Cao:

I'm pleased to inform you that your manuscript has been deemed suitable for publication in PLOS ONE. Congratulations! Your manuscript is now with our production department. 

Kind regards, 

on behalf of

Dr. Muhammad Sajid Hamid Akash 

Academic Editor

PLOS ONE